# Outcome of Patients with Multiple Intracranial Aneurysms after Subarachnoid Hemorrhage and Future Risk of Rupture of Unruptured Aneurysm

**DOI:** 10.3390/jcm10081712

**Published:** 2021-04-15

**Authors:** Seppo Juvela

**Affiliations:** Department of Clinical Neurosciences, University of Helsinki, FI-00029 Helsinki, Finland; seppo.juvela@helsinki.fi; Tel.: +358-50-5457258

**Keywords:** unruptured intracranial aneurysm, natural history, cigarette smoking, outcome, risk factors, subarachnoid hemorrhage

## Abstract

The purpose was to study the risk of rupture of unruptured intracranial aneurysms (UIAs) of patients with multiple intracranial aneurysms after subarachnoid hemorrhage (SAH), in a long-term follow-up study, from variables known at baseline. Future rupture risk was compared in relation to outcome after SAH. The series consists of 131 patients with 166 UIAs and 2854 person-years of follow-up between diagnosis of UIA and its rupture, death or the last follow-up contact. These were diagnosed before 1979, when UIAs were not treated in our country. Those patients with a moderate or severe disability after SAH, according to the Glasgow Outcome Scale, had lower rupture rates of UIA than those with a good recovery or minimal disability (4/37 or 11%, annual UIA rupture rate of 0.5% (95% confidence interval (CI) 0.1–1.3%) during 769 follow-up years vs. 27/94 or 29%, 1.3% (95% CI 0.9–1.9%) during 2085 years). Those with a moderate or severe disability differed from others by their older age. Those with a moderate or severe disability tended to have a decreased cumulative rate of aneurysm rupture (log rank test, *p* = 0.066) and lower relative risk of UIA rupture (hazard ratio 0.39, 95% CI 0.14–1.11, *p* = 0.077). Multivariable hazard ratios showed at least similar results, suggesting that confounding factors did not have a significant effect on the results. The results of this study without treatment selection of UIAs suggest that patients with a moderate or severe disability after SAH have a relatively low risk of rupture of UIAs. Their lower treatment indication may also be supported by their known higher treatment risks.

## 1. Introduction

Subarachnoid hemorrhage (SAH) is a disease with high rates of death and poor outcome [1,2,3]. The rate of treatment of unruptured aneurysms (UIAs) is thus increasing with a goal of reducing incidence of aneurysm rupture [3,4,5]. The effect of treating UIAs on the incidence has been minor [6], however, as compared to that of reduced smoking rates [6,7]. The majority of diagnosed UIAs never rupture during the remainder of the patient’s life-time [8,9].

Since the frequency of aneurysms is not reducing [10], the risk of rupture is probably reducing, resulting in smaller SAH incidence rates [6,7]. The indications for treatment of UIAs are demanding because there are of only a few existing prospective studies of the natural history of UIAs [11]. Such studies are not possible to complete nowadays without treatment selection bias. Most natural history studies have also been derived from patient populations, with treatment selection leading to low rupture rates.

The best-known form of UIA rupture scoring is based on a large meta-analysis of individual data from 8382 patients gathered from six prospective cohort studies [11]. Most patients of these cohorts were those patients who were left for conservative follow-up due to an estimated low rupture risk or high treatment risks. The PHASES score showed the following factors to predict an increased risk of aneurysm rupture during a mean follow-up of 3.5 years: Population, Hypertension, Age and Size of the aneurysm; Earlier aneurysm rupture; and Site of the aneurysm [11]. Juvela score was based on a cohort representing an almost lifelong prospective follow-up of UIAs among those of working age without treatment selection bias [1,8,9,12,13,14]. This score showed the following factors to predict an increased rupture risk: age < 40 years, current smoking, and UIA location and size [15,16]. The same factors also increased the growth risk of UIAs [14,17,18]. Location of UIAs in the anterior communicating artery (ACOM) and posterior communicating artery (PCOM) and the maximum size of UIA were included in both scoring systems, and these also yielded the highest score points.

Previously UIAs were mostly diagnosed in patients with SAH and multiple aneurysms among patients of working age. Increasing use of magnetic resonance imaging for dealing with symptoms unrelated to UIAs (vascular diseases, prolonged headache, dizziness, dementia, etc.) has led to the detection of more incidental UIAs in older patients than previously [3,4,5].

After SAH, the outcome may be worse, e.g., for higher patient age or more frequent re-bleeding episodes in patients with multiple aneurysms than in those with a single aneurysm [3,19]. It is not known whether rupture risk of UIAs in patients with multiple aneurysms after treatment of the ruptured one is different according to outcome after SAH. Treatment risks in those patients with disability may also be higher than in those with a good recovery [4,5].

This cohort represents an almost lifelong prospective follow-up study of UIAs among patients of working age without treatment selection bias [1,8,9,12,13,14,17,18], and it was considered to be of a high quality among all the known UIA series [11,20]. The aim was to study whether future rupture risk of UIAs differs according to the outcome status after SAH and treatment of the ruptured one.

## 2. Methods

### 2.1. Patient Population

The original cohort comprised 142 European white patients (median age 42 years; 76 women) with 182 UIAs diagnosed between 1956 and 1978 at the Department of Neurosurgery, Helsinki University Central Hospital, i.e., at the time when UIAs were not operated on in Finland. This hospital was responsible for neurosurgical services for nearly the whole Finnish population. For details of the cohort with the inclusion and exclusion criteria, as well as follow-up arrangements, see the previous reports [1,8,9,12,13,14,15,16,17,18,21].

Of the 142 patients with UIAs, 131 had a prior SAH (median 42 years; 68 women) with multiple UIAs (*n* = 166) at baseline and were included in this study. Only the verified ruptured aneurysm was operated on, and the occlusion of the ruptured aneurysm without sacrifice of the parent vessel was confirmed by postoperative angiography. Patients who died within 3 months after SAH were excluded from the follow-up study. Outcome after SAH was assessed at 3 months according to the Glasgow Outcome Scale (GOS) [22].

### 2.2. Follow-Up Methods

Detailed follow-up procedures have been reported previously [1,8,9,12,13,14,15,16,17,18,21]. Briefly, the follow-up was based on visits to outpatient clinics, postal questionnaires and telephone interviews obtained from patients and family members approximately every 10 years since the early 1960s. The structured questionnaire included patient characteristics, previous diseases, hospital visits, medication and health behavior. The last follow-up took place in 2012, when 20 patients were still alive without rupture of the UIA.

Additional information on all the patients was obtained from medical records from other hospitals and general practitioners to provide the accuracy of the medical data including blood pressure (BP) values. Autopsy reports and official death certificates were examined for all deceased patients. In Finland, a statutory medico-legal autopsy is performed on all those who die because of trauma or unknown causes (Act on Inquests into the Cause of Death, 459/1973, Finnish Law). The follow-up was complete [1,8,9,12,13,14,15,16,17,18,21].

### 2.3. Risk Factors

Core and most highly recommended supplementary variables for the study of UIA were available for the present purpose [23]. The PHASES [11] and Juvela [15,16] scores were recorded for each patient from variables at the baseline. Hypertension was defined as a systolic pressure repeatedly >140 mm Hg, a diastolic pressure >90 mm Hg at baseline or use of antihypertensive medication [8,13]. Blood pressure (BP) values were recorded before diagnosis and during the first year after SAH, excluding the values obtained within 3 months after SAH, because SAH may secondarily increase BP [12].

Cigarette smoking was grouped as follows: never a smoker, formerly a regular cigarette smoker (quit before or during the follow-up) or currently a cigarette smoker at the end of follow-up. Alcohol consumption was calculated as approximate grams of absolute ethanol consumed within one week (1 standard drink = 12 g of alcohol). A family history of SAH was defined as ≥2 first-degree relatives with verified ruptured aneurysms [8,13].

All angiographies performed at baseline were re-examined by an experienced neuroradiologist [8,12,13]. He had no knowledge of the patients’ case histories. The locations and maximum diameters of the UIAs were measured from standard projections of conventional angiograms at baseline.

### 2.4. Statistical Analysis

The data were analyzed with IBM SPSS Statistics, version 27.0, for Windows (IBM Corp., Armonk, NY, USA). Categorical variables were compared by using Fisher’s exact 2-tailed test or the Pearson Chi-square test, whereas the continuous variables (expressed as mean ± standard deviation (SD) or median with interquartile range (IQR) and/or ranges) were compared by means of the Mann–Whitney U-test, t-tests or Spearman rank correlation coefficients.

Each patient was followed up until SAH, death from a reason other than SAH, occlusion treatment of the UIA (three cases with a follow-up lasting >24.4 years) or the last follow-up contact by time point which came first. The average annual incidence of SAH was calculated by dividing the number of first events of SAH from the index UIA by the number of person-years in the follow-up. Cumulative rates of SAH were estimated by the Kaplan–Meier product-limit method, and the curves for the groups were compared by using the log-rank test.

Cox proportional hazards regression analysis was used to test whether the outcome after prior SAH is an independent predictor of future rupture of diagnosed UIA, taking into account established risk and confounding factors. Wald statistics were employed to estimate hazard ratios (HRs) and 95% confidence intervals (CIs). The proportionality assumption was confirmed. A two-tailed *p*-value < 0.05 was considered statistically significant.

## 3. Results

### 3.1. Patient Characteristics and Follow-Up

During a total follow-up of 2854 person-years (median 21.2, IQR 10.7–32.2 years per patient), 31 of the 131 patients (24%) had an aneurysm rupture, and of them, 14 were fatal. The cumulative rate of SAH was 9% at 10 years and 30% at 30 years. The median follow-up time between the diagnosis of UIA and a subsequent aneurysm rupture was 10.8 years (IQR 8.4–17.4, range 1.2–24.2 years), and the median follow-up for patients without a rupture was 24.2 years (IQR 15.6–35.1, range 0.8–52.3 years).

### 3.2. Outcome after Prior SAH and Subsequent UIA Rupture

After SAH, 94 patients had a good recovery or minimal disability according to GOS, and of them, 27 (29%) had an aneurysm rupture during 2085 person-years of follow-up (approximate annual rupture rate 1.3%, 95% CI 0.9–1.9%). Of 26 patients who had a moderate disability, three (12%) had an aneurysm rupture during 617 person-years (rupture rate 0.5%/year). Only 11 patients were severe disabled, and of them, one (9%) had an aneurysm rupture during 152 person-years (rupture rate 0.7%/year). Since there were only a few patients with severe disability, and because follow-up times did not differ significantly by outcome status, patients with either a moderate or severe disability were combined in subsequent analyses (annual UIA rupture rate, 0.5%; 95% CI, 0.1–1.3%). The cumulative rate of aneurysm rupture was lower (log rank test, *p* = 0.066) in those with a moderate or severe disability than in those with a good recovery or minimal disability (3% (95% CI, 0–10%) vs. 11% (95% CI, 5–18%) at 10 years and 17% (95%CI, 1–32%) vs. 35% (95% CI, 24–46%) at 30 years) (Figure 1).

Patients with a moderate or severe disability were significantly older, and they had non-significantly higher BP values than others (*p* = 0.060–0.072) (Table 1). Location and size of aneurysms, as well as PHASES score, did not differ significantly between outcome groups. Treatment scores (Juvela scores) in which the patients of this cohort also were initially included for estimation [15,16] were significantly (*p* = 0.015) higher in those with a good recovery or minimal disability than in others. Contrary to PHASES score, patients aged <40 years in this score had a higher scoring for risk of rupture. Patients with a good recovery of minimal disability were more frequently alive or had SAH at the end of the follow-up than those with a worse outcome who, on the other hand, had died more often from unrelated causes (Table 1).

### 3.3. Risk Factors for Aneurysm Rupture

Risk factors for aneurysm rupture are shown in Table 2. As expected, rupture risk scores per unit were significant predictors for rupture. Reduced risk of aneurysm rupture of patients with a moderate or severe disability did not reach significance (*p* = 0.077) in univariable analysis. When the outcome after prior SAH was adjusted for PHASES score, sex and hypertension, its HR was almost significant (HR, 0.35; 95% CI, 0.12–1.02; *p* = 0.054). Adjusted HR of moderate or severe disability for UIA rupture was slightly less significant (*p* = 0.14) after adjustment for sex, hypertension and Juvela score; the score also shows a lower rupture risk in older patients.

## 4. Discussion

Both scores of rupture risk of UIAs obtained from this prospective study with an almost lifelong follow-up and low treatment selection bias were significant. Patients with a prior SAH and multiple aneurysms after SAH had an approximately 60% lower annual risk of UIA rupture if patients had moderate or severe disability. This difference showed borderline significance likely because of a small number of patients with a moderate or severe disability. Since this significance further decreased after adjustment for Juvela score, the reason for lower rupture risk in disabled patients was at least partly due to their older age. These patients also died more commonly of causes unrelated to unruptured aneurysms.

The purpose of scoring is to estimate future lifelong rupture risk of only verified UIAs. It is not appropriate to be used for retrospectively scoring aneurysms which have already ruptured. Of ruptured aneurysms, >90% are <10 mm in diameter and 75% are <7 mm [3]. These small aneurysms would yield low scores. Most rupturing small aneurysms rupture likely soon after their appearance because of a high relative growth rate of aneurysm. Relative aneurysm growth rate, taking into account the initial diameter, has been shown to be correlated more highly with aneurysm rupture rate than aneurysm growth rate itself [17].

Location of UIAs in the ACOM and PCOM, and the maximum diameter of UIA, are included in both scoring systems with also the highest score points [11,15,16]. These locations are known to carry a higher rupture risk than others, and the risk may be even higher than has previously been shown in prospective studies because these aneurysms have also been treated more frequently at baseline than others. Both risk scores also predicted long-term UIA ruptures better than did any of the separate factors alone in the scorings [15,16].

Cigarette smoking, patient age, female sex and hypertension have also been known to increase the risk of SAH and possibly rupture of diagnosed UIA. Of these factors, smoking and young patient age are known to increase the risk of rupture of existing aneurysms, while the roles of female sex and hypertension, as well as of prior SAH, aneurysm multiplicity, and family history of aneurysms, are inconsistent, and their impact may be more evident in the risk of aneurysm formation.

Case fatality and poor outcome after aneurysmal SAH are principally determined by clinical and radiological severity of bleeding, and by several laboratory markers, most of which also correlate with the severity grade [3]. The outcome is also independently predicted pre-SAH factors like patient age, hypertension, alcohol consumption, and aneurysm size and location [1,3,24]. Patients with a moderate or severe disability were expectedly significantly older and had almost significantly higher BP values than others. Reduced rupture risk of UIAs of patients with disability remained similar after adjustment for these factors and rupture risk scores. This reduced risk was even near significant after adjustment for PHASES score (*p* = 0.054). The reduced rupture rate was partly explained by a higher patient age among those with a moderate or severe disability. They, on the other hand, had higher BP values and prevalence of hypertension which increase rupture risk. The variables of this study could not explain why these disabled patients had a 60% lower rupture rate. An additional mechanism of action may be the lower physical activity of disabled patients, as an example. Physical activity is known to increase the risk of aneurysm rupture. Another reason may be the fact that disabled patients die more commonly of unrelated causes. The lower rupture risk is important for definition of treatment indication. The patients with a moderate or severe disability are perhaps better to remain in conservative treatment. In a similar way, conservative follow-up is better for the patients with increased mortality due to other reasons; e.g., heavy alcohol drinkers of working age have an annual mortality of 5%, which is clearly higher than UIA rupture risk itself [25]. Patients with a moderate or severe disability also have cerebral infarcts caused by an acute and delayed cerebral ischemia after SAH. Cerebral infarcts also increase the risks of treatment of UIAs and are associated with worse treatment results [3,5,26]. This also speaks for the conservative follow-up of patients with a clear disability.

The strengths of this study are the complete and almost lifelong follow-up of patient of working age and the very limited treatment selection bias [8,9,15,17]. Correspondingly, a previous UIA study based on this cohort was considered to be of high study quality and have low sources of bias relative to other UIA studies [11,20]. A major strength and advantage of this cohort is also the simple treatment scoring with only four variables which is easy and simple to use in everyday clinical practice [15,16,18]. Although Finns have been considered to have a higher risk for SAH, its incidence is not higher when the study design with the inclusion and exclusion criteria, the accuracy of diagnosis, and the sex and age distributions of the population are carefully taken into account [7]. The populations in Nordic countries have similar incidences rates.

One limitation of this study is the relatively small sample size, despite significantly longer follow-up time per patient, as compared with other large prospective studies, which, on the other hand, had high treatment selection bias. This patient population deals with UIAs in cases with multiple aneurysms with treated ruptured ones among those of working age. These patients now represent a minority of UIA cases, but they usually have been considered to be those whose UIAs should be occluded. Patients with a previous SAH or multiple UIAs have not been shown to have an elevated risk of aneurysm rupture when confounding factors are taken into account [11]. This study suggests that UIAs of patients of working age with a prior SAH may have a future rupture risk from UIAs of 60% lower than others if they have moderate or severe disability after SAH. Since these patients are also known to have higher risks and poorer results after treatment of UIAs, they may be more suitable for conservative follow-up unless they have a high rupture risk score, e.g., large ACOM aneurysm among a young smoking patient.

## Figures and Tables

**Figure 1 jcm-10-01712-f001:**
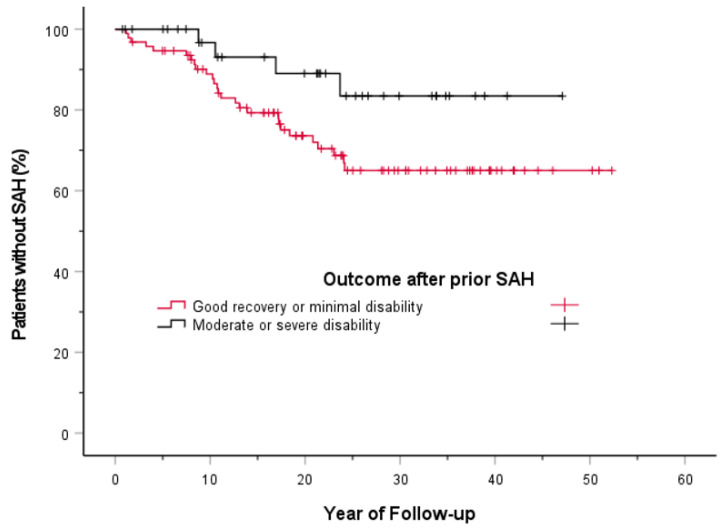
Long-term cumulative aneurysm rupture risk according to outcome after prior subarachnoid hemorrhage (SAH). The markers on the curves indicate censored events. SAH indicates subarachnoid hemorrhage. See text for detailed statistics; *p* = 0.066 for the difference between cumulative aneurysm rupture rates between groups (log-rank test).

**Table 1 jcm-10-01712-t001:** Baseline characteristics, follow-up status and outcome after prior SAH.

Factor	Patients with Good Recovery or Minimal Disability (*n* = 94, 72%)	Patients with Moderate or Severe Disability (*n* = 37, 28%)	All Patients (*n* = 131, 100%)
**Sex (%)**			
Men	46 (73)	17 (27)	63 (100)
Women	48 (71)	20 (29)	68 (100)
**Age**			
Mean (SD) years *	40.1 (9.9)	44.8 (9.3)	41.4 (9.9)
<40 years (%) †	47 (87)	7 (13)	54 (100)
**Current cigarette smoking at baseline (%)**			
No	32 (68)	15 (32)	47 (100
Yes	54 (79)	14 (21)	68 (100)
**Current cigarette smoking at end of follow-up (%)**			
No	40 (70)	17 (30)	57 (100)
Yes	46 (79)	12 (21)	58 (100)
**Hypertension (%)**			
No	63 (76)	20 (24)	83 (100)
Yes	31 (65)	17 (35)	48 (100)
**Blood pressure (mm Hg)**			
Mean (SD)	138 (19)/84 (10)	144 (19)/88 (9)	140 (19)/85 (10)
**Alcohol consumption, *n* = 89**			
≥300 g/week (%)	14 (73)	5 (26)	19 (100)
**Family history, *n* = 88 (%)**	6 (67)	3 (33)	9 (100)
**Location of the largest aneurysm (%)**			
ACOM	5 (71)	2 (29)	7 (100)
PCOM	24 (77)	7 (23)	31 (100)
MCA	41 (66)	21(34)	62 (100)
Other	24 (77)	7 (23)	31 (100)
**Size of the largest aneurysm**			
Mean (SD) mm	4.4 (2.1)	4.4 (2.1)	4.4 (2.1)
Median (IQR) mm	4.0 (3.0–5.0)	4.0 (3.0–5.0)	4.0 (3.0–5.0)
>6 mm	14 (74)	5 (26)	19 (100)
**PHASES score**			
Mean (SD)	4.3 (2.0)	4.4 (1.9)	4.3 (1.9)
Median (IQR) mm	4.0 (3.0–5.3)	4.0 (6.0)	4.0 (3.0–6.0)
**Juvela score**			
Mean (SD) *	3.8 (2.4)	2.6 (2.1)	3.5 (2.4)
Median (IQR) mm	4.0 (2.0–6.0)	2.0 (1.0–4.0)	4.0 (2.0–5.0)
**Status at the end of follow-up (%) ***			
Alive	20 (83)	4 (17)	24 (100)
SAH	15 (88)	2 (12)	17 (100)
Fatal SAH	12 (86)	2 (14)	14 (100)
Died of unrelated causes	47 (62)	29 (38)	76 (100)

Abbreviations: SAH = subarachnoid hemorrhage; SD = standard deviation; IQR = interquartile range (range between the 25th and 75th percentiles); ACOM = anterior communicating artery; MCA = middle cerebral artery; PCOM = posterior communicating artery. Current smoking value was missing for 16 patients. PHASES score groups are shown without Finnish population points (5 points), which had no effect on significance levels. PHASES score: hypertension (1 point); age > 70 years (1 point); aneurysm size 7.0–9.9 mm (3 points), 10.0–19.9 mm (6 points), or ≥20 mm (10 points); earlier SAH (1 point); and site of aneurysm in MCA (2 points), or in ACOM, anterior cerebral artery, PCOM or posterior circulation (4 points for each site). Juvela score: age < 40 years (2 points); current smoking (2 points); aneurysm size > 6 mm (3 points); and aneurysm location in PCOM (2 points), bifurcation of internal carotid artery (4 points) or ACOM (5 points). * *p* < 0.05, † *p* < 0.01.

**Table 2 jcm-10-01712-t002:** Risk factors for the rupture of unruptured intracranial aneurysms.

Factor	Univariable HR (95% CI)	Multivariable HR (95% CI)	
		Model I	Model II
**Sex**			
Men	1.00	1.00	1.00
Women	1.44 (0.69–3.01)	1.40 (0.67–2.94)	1.29 (0.59–2.83)
**Hypertension**			
No	1.00	1.00	1.00
Yes	1.46 (0.72–2.99)	1.45 (0.70–3.00)	2.08 (0.94–4.60)
**Outcome at 3 months after prior SAH**			
good recovery or minimal disability	1.00	1.00	1.00
moderate or severe disability	0.39 (0.14–1.11)	0.35 (0.12–1.02)	0.39 (0.11–1.37)
**PHASES score per unit**	1.25 (1.05–1.49) *	1.24 (1.03–1.48) *	
**Juvela score per unit**	1.46 (1.25–1.70) †		1.47 (1.25–1.72) †

Abbreviations: SAH = subarachnoid hemorrhage; HR = hazard ratio; CI = confidence interval. In the multivariable models, the HRs were adjusted for the other variables listed in the column. * *p* < 0.05, † *p* < 0.01.

## Data Availability

Anonymized data not published within this article are available by request from any qualified investigator.

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
