# Peer review of "Outcome of Patients with Multiple Intracranial Aneurysms after Subarachnoid Hemorrhage and Future Risk of Rupture of Unruptured Aneurysm"

_jcm, 2021, doi:10.3390/jcm10081712_

Round 1

Reviewer 1 Report

I would now accept the paper.

Author Response

Reviewer´s comment:

I would now accept the paper.

Author’s response:

I thank reviewer for this comment. No additional revision were recommended.

Reviewer 2 Report

I would like to thanks the authors for adding some of recommandations i've provided.

However, in my opinion the conclusion is still not supported by results.

Author Response

Reviewer´s comment:

I would like to thanks the authors for adding some of recommandations i've provided.

However, in my opinion the conclusion is still not supported by results.

Author’s response:

I thank the reviewer for the comments of this paper. Patients with moderate and severe disability after SAH have lower rupture risk of their UIAs as compared with those with a better outcome.

I have now changed conclusions. The last sentence of Abstract is as follows: “Their lower treatment indication may also be supported by their known higher treatment risks.”

The last sentence of the Discussion is now as follows: “Since these patients are also known to have higher risks and poorer results after treatment of UIAs they may be more suitable for conservative follow-up unless they have a high rupture risk score e.g., large ACOM aneurysm among a young smoking patient.”

Reviewer 3 Report

Dear Editor, 
Dear Dr. Juvela,

I have read with an interest the original paper of Juvela. The author took up a very current and important topic of the outcome of patients with multiple intracranial aneurysms after subarachnoid hemorrhage and future risk of rupture of unruptured aneurysm. In my opinion, the manuscript in this shape requires some revision. The following issues need correction:

Major:

  • "Those with a moderate or severe disability had a decreased cumulative rate of aneurysm rupture  (log  rank  test,  p=0.066) and risk  of  UIA  rupture  (hazard  ratio  0.39,  95% CIl  0.14-1.11, p=0.07" - no they did not have "a decreased (...) rate". Provided p>0.05 means that there were not any significant differences and both values were at comparable level! It must be rewritten.
  • What with the treatment approach of an unruptured aneurysm? The Journal of Clinical Medicine have published an interesting paper of 3d printing in intracranial management (see: https://www.mdpi.com/2077-0383/10/6/1201) - these results should be discussed in the current manuscript.

Minor:

  • Ln 19: 95% CI instead of 95% CII
  • Ln 119: in the manuscript median value is provided with both IQR and range. It should be stated in this section.
  • Figure 1: the p-value of log rank test should be provided at the Figure 1
  • Table 1: p-values should be provided in the separate columns

Good job!

Author Response

Reviewer´s comment:

I have read with an interest the original paper of Juvela. The author took up a very current and important topic of the outcome of patients with multiple intracranial aneurysms after subarachnoid hemorrhage and future risk of rupture of unruptured aneurysm. In my opinion, the manuscript in this shape requires some revision. The following issues need correction:

Author’s response:

I thank reviewer for these comments dealing with importance of this work.

Reviewer´s comment:

Major:

  • "Those with a moderate or severe disability had a decreased cumulative rate of aneurysm rupture  (log  rank  test,  p=0.066) and risk  of  UIA  rupture  (hazard  ratio  0.39,  95% CIl  0.14-1.11, p=0.07" - no they did not have "a decreased (...) rate". Provided p>0.05 means that there were not any significant differences and both values were at comparable level! It must be rewritten.

Author’s response:

Log rank test compares the absolute (cumulative) changes between the groups while Cox models the relative changes. Both these tests are important. The previous one is more important to estimate real risk in practice while the latter is purposeful for multivariable modeling. Both these test were just above level of p=0.05. So, I changed the text as follows in the Abstract: “Those with a moderate or severe disability tended to have a decreased cumulative rate of aneurysm rupture (log rank test, p=0.066) and lower relative risk of UIA rupture (hazard ratio 0.39, 95% CI 0.14-1.11, p=0.077)”.

Reviewer´s comment:

What with the treatment approach of an unruptured aneurysm? The Journal of Clinical Medicine have published an interesting paper of 3d printing in intracranial management (see: https://www.mdpi.com/2077-0383/10/6/1201) - these results should be discussed in the current manuscript.

Author’s response:

It remained obscure to me how the recently published paper in JCM is beneficial for the discussion of current paper. The current paper deals with risk and risk factors of UIA rupture which is crucial for treatment decisions. The recently published paper is a technical paper for planning of intracranial aneurysms in general (both ruptured and unruptured ones). Although I tried to estimate how I discuss from the results of that paper in my paper, I found no relevant connections between the results of these papers. I likely causes conclusions of the current paper even to be less clear. Was the reviewer of this paper also an author of that paper? Suggested citation: “BÅ‚aszczyk, M.; Jabbar, R.; Szmyd, B.; Radek, M. 3D Printing of Rapid, Low-Cost and Patient-Specific Models of Brain Vasculature for Use in Preoperative Planning in Clipping of Intracranial Aneurysms. J. Clin. Med. 2021, 10, 1201. https://doi.org/10.3390/jcm10061201”

Reviewer´s comment:

Minor:

  • Ln 19: 95% CI instead of 95% CII

Author’s response:

I thank the reviewer for this typo. I have now corrected this.

Reviewer´s comment:

  • Ln 119: in the manuscript median value is provided with both IQR and range. It should be stated in this section.

Author’s response:

I thank the reviewer for my missing of term “ranges” from the Methods (Statistical analysis section). I have now added it after IQR.

Reviewer´s comment:

  • Figure 1: the p-value of log rank test should be provided at the Figure 1

Author’s response:

Detailed statistics are in the text just above the Figure. I have now also added log rank test the figure text: “P= 0.066 for the difference between cumulative aneurysm rupture rates between groups (log rank test)”.

Reviewer´s comment:

  • Table 1: p-values should be provided in the separate columns

Author’s response:

There were only 3 significant difference in comparisons between outcome group (age, Juvela score and status at the end of follow-up). Adding of column with numerous nonsignificant p- values are not reasonable and makes the table less clear. I have used p- levels after significant variables (*p<0.05, † p<0.01) which are also shown now more clearly in the footnote.

Reviewer´s comment:

Good job!

Author’s response:

I thank reviewer for this comment.

Round 2

Reviewer 3 Report

I would like to thank you very much for provided changes. In my opinion, the paper may be accepted in the current form.

This manuscript is a resubmission of an earlier submission. The following is a list of the peer review reports and author responses from that submission.

Round 1

Reviewer 1 Report

This is a very interesting and well written work. The historical and well recognized population is a very valuable material.

However I have concerns about the conclusions drawn from the results.

Results are not clearly presented; Kaplan maier curves could be displayed.

Adding PHASE and Juvela’s score seems confusing here.

Even, if one should not focused on “P value”, that’s a fact, these are not significant in both analysis here.

Indeed; the population with severe and moderate disability is not only much smaller than the good recovery population but this population is also followed-up during a much shorter time (of course).

Then, the paucity of events during this time results in very large Confidence intervals that , at last, end-up with non-significant p values and HRs that encompass the value 1.

Hence, in my opinion there is no reason to claim that moderate and severely disable patients “ are better suitable for conservative follow-up” .

Author Response

RESPONSES TO REVIEWERS

I would like to thank both reviewers for their encouraging comments and swift review of this paper. It is evident, that the reviewers have been very careful and spent time to improve and check the quality of this manuscript. The useful comments and few available suggestions were helpful for revising the manuscript.

I hope that the revised manuscript and the responses to reviewers shown below could lead to an elevation of publication priority score.

My point-by-point responses to the reviewers' comments are given below.

Responses to Reviewer 1

Reviewer´s comment:

This is a very interesting and well written work. The historical and well recognized population is a very valuable material.

Author’s response:

I thank reviewer for this comment. The reason is that this cohort consists of patients with UIAs without treatment selection bias and almost lifelong follow-up. This kind of study is no more possible to perform.

Reviewer´s comment:

However I have concerns about the conclusions drawn from the results.

Results are not clearly presented; Kaplan maier curves could be displayed.

Author’s response:

I have now added Kaplan–Meier curves (Figure 1). Previously, cumulative rupture rates and statistics were also already previously shown in text. Proportionality of curves by groups can be seen.

Reviewer´s comment:

Adding PHASE and Juvela’s score seems confusing here. Even, if one should not focused on “P value”, that’s a fact, these are not significant in both analysis here.

Author’s response:

Scores in the analyses are useful since these consist simultaneously of only significant risk factors making results more reliable than using different risk factors with multiple comparisons when Bonferroni type corrections should be done. See also responses to reviewer 2, who thought these to be valuable. Comparing new risk factors with several esteemed scores makes the new results also more valid. After adjustment HR of outcome for scores changes only little significance levels and remained almost significant. Unfortunately, larger patient population is no more available for unbiased study.

Reviewer´s comment:

Indeed; the population with severe and moderate disability is not only much smaller than the good recovery population but this population is also followed-up during a much shorter time (of course).

Author’s response:

In fact, follow-up times were not significantly different between outcome groups (Results paragraph 3.2 and Figure 1). The follow-up terminated more frequently to aneurysm rupture (down steps in curves) among patients with good recovery or minimal disability and to unrelated causes of death among those with more impaired outcome (markers on the curves). In Figure 1 and Table 1, it can also be seen that censored events (unrelated causes of death) are also common in those with a favorable outcome.

Reviewer´s comment:

Then, the paucity of events during this time results in very large Confidence intervals that , at last, end-up with non-significant p values and HRs that encompass the value 1.

Author’s response:

I have added now also 95% CI both to annual rupture rates and cumulative rupture rates at 10 and 30 years after the diagnosis. These percentages show absolute rupture rates where differences between outcome groups are easier to see than observing only relative differences between groups (HRs with 95% CIs). Both annual and cumulative rupture rates of patients with more impaired outcome are lower than are known risks of treatment of these patients. Furthermore cumulative rupture rates should be discounted to time point of possible treatment (e.g. by 5% per year). Treatment risks occur at once after aneurysm occlusion but in the conservative follow-up ruptures occur linearly over decades, not at the time of diagnosis.

Reviewer´s comment:

Hence, in my opinion there is no reason to claim that moderate and severely disable patients “ are better suitable for conservative follow-up” .

Author’s response:

This study is a scientific one based on a reliable data, the results of which I have interpreted as can do every reader, of course, after seeing these results. I have participated also as principal investigator both in PHASES (in reference list, ref. 11) and in UIATS (ref. 5) studies, of which the latter was based purely on opinions of treating doctors. The results of UIATS were modest (ref. 15) while those of PHASES were better (ref. 16).

Reviewer 2 Report

The authors conducted a prospective life-long study in a small cohort of patients with multiple intracranial aneurysms that initially presented with subarachnoid hemorrhage. The authors concluded that patients that have moderate to severe disability have lower risk of aneurysm rupture compared to patients that recovered well from the initial SAH, and that disabled patients may benefit from a more conservative approach to their unruptured aneurysm. The authors also apply the Juvela score and compare it the PHASES score to define its predictive value of rupture risk in this unique cohort, in addition to other clinical factors. They also apply these scores and clinical factors in comparing initial outcomes for these patients. 

The manuscript is well written, and this cohort of patients and the data presented is extremely beneficial to physicians that take care of patients with brain aneurysms. 

Interestingly patients with higher Juvela scores were more likely to have a good outcome initially which is likely tied to the (<40) age point scoring system. In addition 29/38 patients with moderate to severe disability after SAH died of unrelated causes in long-term follow up. This is compared to 47/88 patients with initial good outcomes dying of unrelated causes. 

Since age was greatest predictor of poor outcomes after initial SAH one could argue that this rupture risk for the other unruptured aneurysm is ultimately tied to just age of the patient and not outcome. Therefore the risk of an unruptured aneurysm remains the same year to year regardless of initial outcome as long as you're alive to realize that risk. When one is old and dying of unrelated causes that risk cannot be realized. 

Another comment is in regards to the methodology of determining which aneurysm ruptured initially. In your referenced article you had 5 patients with lateralizing symptoms, greater size in 10 patients, size and secondary sac in 69 patients, and ICH in 38 patients; therefore in a large percentage of patients this method using size +/- secondary sac (79) this was an instinctual determination. This is certainly neurosurgical dogma, however I think over the decades we have learned that there is probably more to determining rupture site than just size, and secondary sac. It would be helpful to know the timing of second SAH compared to the first to rule out potential judgement errors in determining the actual initial aneurysm rupture location. In my experience one cannot always know which aneurysm ruptured in patients presenting with SAH and having multiple intracranial aneurysms. If that occurs in my practice I usually advocate for treatment of all the suspect aneurysms initially, if it can be done safely.

Last comment is in regards to the patient population which is as described homogenous. Therefore when making conclusions in regards to these findings this should be included in the limitations. 

Author Response

RESPONSES TO REVIEWERS

I would like to thank both reviewers for their encouraging comments and swift review of this paper. It is evident, that the reviewers have been very careful and spent time to improve and check the quality of this manuscript. The useful comments and few available suggestions were helpful for revising the manuscript.

I hope that the revised manuscript and the responses to reviewers shown below could lead to an elevation of publication priority score.

My point-by-point responses to the reviewers' comments are given below.

Responses to Reviewer 2

Reviewer´s comment:

The authors conducted a prospective life-long study in a small cohort of patients with multiple intracranial aneurysms that initially presented with subarachnoid hemorrhage. The authors concluded that patients that have moderate to severe disability have lower risk of aneurysm rupture compared to patients that recovered well from the initial SAH, and that disabled patients may benefit from a more conservative approach to their unruptured aneurysm. The authors also apply the Juvela score and compare it the PHASES score to define its predictive value of rupture risk in this unique cohort, in addition to other clinical factors. They also apply these scores and clinical factors in comparing initial outcomes for these patients. 

The manuscript is well written, and this cohort of patients and the data presented is extremely beneficial to physicians that take care of patients with brain aneurysms. 

Author’s response:

I thank the reviewer for the highly complimentary and encouraging comments of this paper.

Reviewer´s comment:

Interestingly patients with higher Juvela scores were more likely to have a good outcome initially which is likely tied to the (<40) age point scoring system. In addition 29/38 patients with moderate to severe disability after SAH died of unrelated causes in long-term follow up. This is compared to 47/88 patients with initial good outcomes dying of unrelated causes. 

Author’s response:

In fact, 29 of 37 (78%) patients with severe or moderate disability died of unrelated causes (38% of all cases with unrelated death) (Table 1). Correspondingly, 47 of 94 patients (50%) with a better outcome died of unrelated causes (62% of all cases with unrelated death). As responded to reviewer 1, patients with more impaired outcome died more frequently of unrelated causes while follow-up terminated more likely to aneurysm rupture in those with a better outcome.

Reviewer´s comment:

Another comment is in regards to the methodology of determining which aneurysm ruptured initially. In your referenced article you had 5 patients with lateralizing symptoms, greater size in 10 patients, size and secondary sac in 69 patients, and ICH in 38 patients; therefore in a large percentage of patients this method using size +/- secondary sac (79) this was an instinctual determination. This is certainly neurosurgical dogma, however I think over the decades we have learned that there is probably more to determining rupture site than just size, and secondary sac. It would be helpful to know the timing of second SAH compared to the first to rule out potential judgement errors in determining the actual initial aneurysm rupture location. In my experience one cannot always know which aneurysm ruptured in patients presenting with SAH and having multiple intracranial aneurysms. If that occurs in my practice I usually advocate for treatment of all the suspect aneurysms initially, if it can be done safely.

Author’s response:

In addition to radiological signs of ruptured aneurysms in all cases with multiple aneurysms, we have also used for identification of ruptured aneurysms surgical records. In that paper we noted: “Identification of ruptured aneurysm in all patients with multiple aneurysms was based on a clear statement of signs of rupture in the surgical record (ref. 12). We excluded all patients without such identification. This was done to exclude to as great degree as possible misclassifications. Furthermore we stated: “Patients with symptomatic aneurysms were included in the study only if SAH was excluded by examining the results of a lumbar puncture within a few days after onset of symptoms.” This latter does not deal with this study since all patients were multiple aneurysm cases with one ruptured one.

The results speak for the fact that misclassification was very unlikely since ruptures occurred linearly by time since the start of follow-up. If patients had ruptured one these likely rupture soon after diagnosis since their rupture risk remain higher for 10 years after the diagnosis (highest risk within first 6 months). The first rupture occurred not earlier than 1.2 years after diagnosis (see results paragraph 3.1): “The median follow-up time between the diagnosis of UIA and a subsequent aneurysm rupture was 10.8 years (IQR 8.4-17.4, range 1.2-24.2 years), and the median follow-up for patients without a rupture was 24.2 years (IQR 15.6-35.1, range 0.8-52.3 years).” Thus it is very unlikely that our UIA cohort included at the start of follow-up also ruptured ones. For comparison, in the ISUIA the highest risk aneurysm rupture was within the 1st year after the diagnosis (se ref. 26). So, it is likely that in the ISUIA the exclusion of ruptured aneurysms was not as carefully done as in our cohort both for asymptomatic/symptomatic or multiple aneurysm cases with treated ruptured one.

Kaplan–Meier curves also show that rupture risk was linear between 0-25 years after diagnosis excluding the possibility that ruptures ones were included at start of follow-up (Figure 1). This differs from corresponding curves of the ISUIA.

Reviewer´s comment:

Last comment is in regards to the patient population which is as described homogenous. Therefore when making conclusions in regards to these findings this should be included in the limitations. 

Author’s response:

I have inserted in the limitations of study of Discussion (in the 2nd last paragraph) following sentences:

“This patient population deals with UIAs in cases with multiple aneurysms with treated ruptured one among those with working age. These patients represent now minority of UIA cases, but they usually have been considered to be those, whose UIAs should be occluded.”